# Investigation of Neurodevelopmental Deficits of 22 q11.2 Deletion Syndrome with a Patient-iPSC-Derived Blood–Brain Barrier Model

**DOI:** 10.3390/cells10102576

**Published:** 2021-09-28

**Authors:** Yunfei Li, Yifan Xia, Huixiang Zhu, Eric Luu, Guangyao Huang, Yan Sun, Kevin Sun, Sander Markx, Kam W. Leong, Bin Xu, Bingmei M. Fu

**Affiliations:** 1Department of Biomedical Engineering, The City College of the City University of New York, New York, NY 10031, USA; yli012@citymail.cuny.edu (Y.L.); mail.yifan.xia@gmail.com (Y.X.); ericluu24@gmail.com (E.L.); guangyao0516@gmail.com (G.H.); 2Department of Psychiatry, Columbia University, New York, NY 10032, USA; vian10032@gmail.com (H.Z.); yan.sun@nyspi.columbia.edu (Y.S.); kevin.sun@vanderbilt.edu (K.S.); sm2643@cumc.columbia.edu (S.M.); 3Department of Biomedical Engineering, Columbia University, New York, NY 10027, USA; kam.leong@columbia.edu

**Keywords:** blood–brain barrier, DiGeorge syndrome, endothelial glycocalyx, tight junction, permeability, transcription network

## Abstract

The blood–brain barrier (BBB) is important in the normal functioning of the central nervous system. An altered BBB has been described in various neuropsychiatric disorders, including schizophrenia. However, the cellular and molecular mechanisms of such alterations remain unclear. Here, we investigate if BBB integrity is compromised in 22q11.2 deletion syndrome (also called DiGeorge syndrome), which is one of the validated genetic risk factors for schizophrenia. We utilized a set of human brain microvascular endothelial cells (HBMECs) derived from the induced pluripotent stem cell (iPSC) lines of patients with 22q11.2-deletion-syndrome-associated schizophrenia. We found that the solute permeability of the BBB formed from patient HBMECs increases by ~1.3–1.4-fold, while the trans-endothelial electrical resistance decreases to ~62% of the control values. Correspondingly, tight junction proteins and the endothelial glycocalyx that determine the integrity of the BBB are significantly disrupted. A transcriptome study also suggests that the transcriptional network related to the cell–cell junctions in the compromised BBB is substantially altered. An enrichment analysis further suggests that the genes within the altered gene expression network also contribute to neurodevelopmental disorders. Our findings suggest that neurovascular coupling can be targeted in developing novel therapeutical strategies for the treatment of 22q11.2 deletion syndrome.

## 1. Introduction

Microdeletions at band q11.2 of human chromosome 22 are the most common interstitial deletion in humans, with an incidence of approximately 1 in every 4000 live births [1]. Patients who carry a 22q11.2 microdeletion can present with highly variable phenotypes that involve multiple organ systems, which historically have been categorized and referred to as DiGeorge syndrome, velo-cardio-facial (VCF) syndrome, or 22q11.2 deletion syndrome (22q11.2DS) [1]. About one-third of 22q11.2 deletion carriers develop schizophrenia or schizoaffective disorder, which is roughly 30 times higher than the general population [2]. It is well known that schizophrenia patients require long-term medical care and impose a major economic burden on the healthcare system [3]. To deal with this challenge, it is important to understand the pathophysiology underlying these psychiatric conditions. An important entry point for such investigations is 22q11.2DS. However, the large number of genes that reside within the 22q11.2 microdeletion add significant complexity to studying the molecular mechanisms associated with the observed neuropsychiatric phenotypes. Across the major psychiatric disorders, blood–brain barrier (BBB) dysregulation is one of the widely observed phenotypes [4,5]. For example, abnormalities in neurovascular unit (NVU) components have been reported in the frontal cortex of the postmortem brains of schizophrenia and autism patients [6,7,8,9]. Of a particular note, endothelial tight junction signaling has been shown to be altered in patients with schizophrenia, in which 12 out of 21 tight-junction-related genes are reduced [10]. Several genes within the deletion domain, such as *claudin-5* and *TBX1*, are critical for the integrity of the BBB and vascular development [11,12]. Alterations in the expression of claudin-5 have been observed in the prefrontal cortex of patients with schizophrenia [13]. Therefore, we speculated that BBB integrity is compromised in 22q11.2DS-associated schizophrenia (22q11.2DS-SCZ), which could contribute to psychiatric-disease-related pathophysiology in patients with 22q11.2DS-SCZ.

To protect the brain from blood-borne neural toxins, the BBB between the central nervous system (CNS) and the cerebral circulation maintains very low permeability due to its unique structure [14]. The brain-specific microvascular endothelial cell (BMEC) is an essential structural component of the BBB. The barrier function of the BBB is mainly determined by two anatomic features of BMECs. One is an elaborated paracellular junction composed of tight junctions (TJs) and adherens junctions (AJs) in the narrow cleft between adjacent ECs [15,16,17]. Another is a matrix-like surface glycocalyx layer (endothelial surface glycocalyx, ESG) of a mucopolysaccharide structure consisting of glycoproteins, acidic oligosaccharides, terminal sialic acids, proteoglycan, and glycosaminoglycan aggregates [18,19,20,21]. Together, these structural components confer an immune privilege to the CNS by restricting interactions with the periphery [16,22,23,24]. More specifically, the ESG limits the interaction of the circulating blood cells with the ECs forming the vessel wall, and partially restricts the passage of large molecules, e.g., plasma proteins and cytokines, across the BBB, while the junction proteins mainly restrict the transport of ions and small molecules [14,19,25,26,27].

To test whether the BBB is compromised in 22q11.2DS, we assessed both the function (i.e., permeability and TEER) and structural components (i.e., junction proteins and the ESG) of the induced BBB (iBBB) derived from iPSCs of patients with 22q11.12DS-SCZ. We employed a protocol widely used for human brain microvascular endothelial cell (HBMEC) differentiation from human iPSCs [28,29,30] to generate HBMECs from three patients with 22q11.2DS-SCZ (DEL) and three age- and sex-matched healthy controls (WT). We formed iBBBs out of these HBMECs on a Transwell filter and quantified their permeability to both a small solute (sodium fluorescein) and a large solute (Dex-70K). We also quantified the trans-endothelial electrical resistance (TEER), an indicator for the permeability to ions. Furthermore, we investigated the molecular mechanisms underlying the permeability changes by using immunostaining and RNA sequencing.

## 2. Materials and Methods

### 2.1. hiPSC Generation and Characterization

Two pairs of 22q11.2DS case (DEL1 and DEL2)/control (WT1 and WT2) hiPSC lines were obtained from the NIMH Repository and Genomics Resource (http://www.nimhstemcells.org/, accessed on 9 May 2018) [31]. One pair of hiPSC lines (DEL3 and WT3) were generated at the Columbia Stem Cell Core via non-integrating Sendai virus-based reprogramming [32] of monocytes from a donor with 22q11.12DS-SCZ, as well as from a healthy sibling control. Usage of these iPSC lines was been approved by the Columbia University Embryo and Embryonic Stem Cell Research Committee (ESCRO). To confirm the stemness of the hiPSC lines, we performed RT-PCR for markers *NANOG* and *OCT4/SOX4* (Figure A1A) and immunostaining of stem cell markers SSEA-4 and TRA-1–60 (Figure A1B). Karyotyping was performed on twenty G-banded metaphase cells at 450–500 band resolution, as previously described [33,34], to ensure the absence of chromosomal abnormalities in all patient- and control-derived cell lines (Figure A1C). We confirmed the genotypes of patient- and control-derived hiPSCs using a multiplex ligation-dependent probe amplification (MLPA) assay to detect copy number changes (Figure A1D). The MLPA results further confirmed that the size of the microdeletion in all patients was between 2M to 3M bps.

### 2.2. HBMEC Differentiation

All iPSCs were maintained on Matrigel (Corning, Corning, NY, USA)-coated wells in an mTeSR1 medium (STEMCELL Technologies, Vancouver, BC, Canada). iPSCs were singularized with 10 mM EDTA and were seeded at a density of 3 × 10^4^ cells/cm^2^ in mTeSR1 for 3 days with 10 μM Rho-associated protein kinase (ROCK) inhibitor Y-27632 (Selleckchem, Houston, TX, USA) added at the first day of culture. Differentiation was conducted as described in [29]. Various mesoderm and endothelial cell (EC) markers were checked at different developmental stages to ensure the cell identity. The final differentiated HBMECs were maintained in a human endothelial serum-free medium (hESFM, Thermo Fisher Scientific, Waltham, MA, USA) until downstream analyses were conducted.

### 2.3. Generation of Induced BBB (iBBB) and Trans-Endothelial Electrical Resistance (TEER) Measurement

Transwell inserts (Falcon, Corning, NY, USA) (bottom area of 0.9 cm^2^) with a 0.4 μm pore size transparent PET membrane were first coated with human-plasma-derived fibronectin (100 μg/mL)/human-placenta-derived collagen IV (400 μg/mL) (Sigma-Aldrich St. Louis, MO, USA) and incubated overnight at 37 °C [29,30]. HBMECs were detached with Accutase, filtered through a 30 μm membrane, and seeded at a density ~50k/cm^2^ on the Transwell insert. The cells were kept in a medium composed of hESFM supplemented with 1% platelet-poor plasma-derived serum (PDS, Alfa-Aesar, Ward Hill, MA, USA), 2% B-27 (Gibco, Thermo Fisher Scientific, Waltham, MA, USA), 2 nM RA (retinoic acid) (Sigma-Aldrich), 50U/mL penicillin-streptomycin (Gibco, Thermo Fisher Scientific, Waltham, MA, USA), 20 ng/mL bFGF, and 10 μM Y-27623 dihydrochloride (Tocris Bioscience, Bristol, UK). The HBMEC monolayer reached confluency in 4–5 days and formed an iBBB in 6–7 days. The confluency of the monolayer was checked every other day in the first 4 days by a bright-field microscope after seeding. The TEER of the monolayer was then monitored every day on days 5–7. When the TEER was stable for at least two consecutive days, the iBBB was considered formed. The TEER was measured by an EVOM2 Epithelial Volt/Ohm Meter with STX2 (World Precision Instruments, Sarasota, FL, USA). The TEER of the blank filter was also measured separately as a background control. The background TEER value was subtracted from the (iBBB + blank filter) measurement to obtain the actual TEER values of the iBBB.

### 2.4. Measurement of Solute Permeability (P)

On the day that we measured the solute permeability (P), 1.5 mL of 10 mg/mL bovine serum albumin (BSA, Sigma-Aldrich) in a DMEM/F-12 medium without phenol red (Corning, Corning, NY, USA) was added to the lower chamber of the Transwell filter. The upper chamber was filled with 0.5 mL of either 10 μM FITC–Dextran 70k (Dex-70k, MW 70 kD, Sigma-Aldrich) or 30 μM sodium fluorescein (NaFl, MW 376, Sigma-Aldrich) in the same medium as in the lower chamber. The samples of 50 μL were drawn every 10 min for 90 min from the lower chamber and were then replaced with the same amount of fresh medium. The concentration (intensity) of the collected fluorescent solutions was determined by a SpectraMax M5 microplate reader (Molecule Devices, San Jose, CA, USA). The BBB permeability (*P*) to a solute was calculated by Equation (1):(1)$$P=\frac{\frac{\Delta C_{L}}{\Delta t}\times V_{L}}{C_{U}\times S}$$
where $\frac{\Delta C_{L}}{\Delta t}$ is the increase in fluorescence concentration in the lower chamber during the time interval Δ*t*, $C_{U}$ is the fluorescence concentration in the upper chamber, $V_{L}$ is the medium volume in the lower chamber, and *S* is the area of the filter membrane [26,35]. If *P_t_* is the measured permeability of both the HBMEC monolayer (iBBB) and the transwell insert membrane, *P_i_* is the measured permeability of a blank transwell filter and *P_iBBB_* is the permeability of the iBBB, which was calculated by Equation (2):(2)$$\frac{1}{P_{t}}=\frac{1}{P_{iBBB}}+\frac{1}{P_{i}}$$

### 2.5. Immunostaining of Tight Junction Proteins and the Endothelial Surface Glycocalyx (ESG)

Labeling tight junction proteins (ZO-1 and occludin) [28,29]: The monolayer was washed three times with DPBS, fixed with ice-cold methanol for 15 min, and blocked with 10% normal goat serum (NGS, Jackson Immuno Research, West Grove, PA, USA) in 0.1% Triton X-100 (Sigma-Aldrich) for 1 h in RT. The cells were then incubated in ZO-1 polyclonal antibody (1:100, 40–2200, host: rabbit; Invitrogen, Thermo Fisher Scientific, Waltham, MA, USA), or occludin monoclonal antibody (1:200, OC-3F10, host: mouse; Invitrogen, Thermo Fisher Scientific, Waltham, MA, USA) at 4 °C overnight [28,29]. After being washed three times with DPBS, cells were incubated for 1 h at RT in Alexa-Fluor-488-conjugated anti-rabbit antibody (1:200, 711-546-152, Jackson Immuno Research, West Grove, PA, USA) for labeling ZO-1 or Alexa-Fluor-488-conjugated goat anti-mouse antibody (1:200, A11001, Invitrogen, Thermo Fisher Scientific, Waltham, MA, USA) for labeling occludin [28,29]. The monolayer was washed three times with DPBS, followed by the mounting of DAPI Fluoromount-G (SouthernBiotech, Birmingham, AL, USA) and made into slides for later visualization.

Labeling the ESG (HS): Since heparan sulfate (HS) is the most abundant glycosaminoglycan (GAG) of the ESG, we quantified HS to represent the ESG [17,25,35,36]. The HBMEC monolayer was first washed three times with 10 mg/mL BSA in Dulbecco’s Phosphate-Buffered Saline (DPBS, Corning, Corning, NY, USA) and then fixed with 2% paraformaldehyde (Polyscience, Warrington, PA, USA) and 0.1% glutaraldehyde (Sigma-Aldrich) for 20 min at RT. It was blocked with 2% NGS for 30 min at RT and then incubated with a monoclonal anti-heparan sulfate antibody (1:100, F58-10E4, host: mouse; Amsbio, Abingdon, UK) at 4 °C overnight. After being washed three times with DPBS, the cells were incubated for 1 h at RT with Alexa-Fluor-488-conjugated goat anti-mouse antibody (1:200, A1101 Invitrogen, Thermo Fisher Scientific, Waltham, MA, USA). The HBMEC monolayer was washed three times with DPBS, followed by mounting of DAPI Fluoromount-G and made into slides for later observation.

### 2.6. Confocal Microscopy and Quantification of Tight Junction Proteins and the ESG

All the samples were imaged by a Zeiss LSM 800 confocal laser scanning microscope with a 40×/NA1.3 oil immersion objective lens. For imaging junction proteins, three fields of 160 μm × 160 μm (2048 × 2048) were randomly chosen for each sample, and captured as a z-stack of 50–60 images with a z-step of 0.2 μm for two channels (AF488 and DAPI). For ESG imaging, three fields of 320 μm by 320 μm (2048 × 2048) for each sample were captured as a z-stack of 30–40 images with a z-step of 0.32 μm. Image projection and intensity quantification for the junction proteins and the ESG were performed by Zeiss ZEN and NIH ImageJ [35]. We used our previously developed protocol in [35,37] for the quantification of the junction proteins. Briefly, the averaged intensity profile from the 3–5 perpendicular lines (~3 μm) equally distributed along each junction (EC–EC border) was determined for that junction. Twenty–thirty junctions (20–30 EC pairs) were randomly selected for each view field (160 μm × 160 μm). The background was subtracted from the non-border region in that view field. Three fields of 60–90 junctions were measured for each sample. Three samples of 180–270 junctions were determined for each case. Since we performed all the experiments on the control (WT) and patients (DEL) for each pair simultaneously, we used the averaged peak intensity of the junction protein from the 3 WT samples for the normalization. For quantifying the ESG, the averaged intensity of 3 fields (each field 320 μm × 320 μm) was measured for each sample. Three samples were measured for each case. The average of 3 WT samples was used for the normalization in ESG quantification in each WT-DEL pair.

### 2.7. Total RNA Isolation and Bulk RNA Sequencing

Six iPSC-derived HBMEC samples, i.e., 3 control-derived lines (WT) and 3 22q11.12DS-SCZ-patient-derived lines (DEL) were collected from the Transwell iBBB model after the functional test (TEER) was completed for each sample. Briefly, the cell culture medium was removed and the cell monolayer in the Transwell filter was washed twice with DPBS at RT. Five hundred microliters of QIAzol lysis reagent (Qiagen, Germantown, MD, USA) was added and the cells were thoroughly lysed by pipetting up and down the lysis reagent until there was no cell debris. The lysed cells were transferred to a microcentrifuge tube for each sample. Total RNA was extracted using a miRNeasy kit (Qiagen, Germantown, MD, USA) according to the instruction of the manufacturer. The concentration and purity of each sample were determined by a spectrophotometer (ND1000; Nanodrop, Thermo Fisher Scientific, Waltham, MA USA) and the quality of RNA was confirmed (with RNA integrity index (RIN) score > 9) by microchip gel electrophoresis (Agilent, Santa Clara, CA, USA) using an Agilent 2100 Bioanalyzer Chip according to the instruction of the manufacture. A poly-A pull-down step was performed to enrich mRNAs from total RNA samples, and library preparation was embarked upon by using an Illumina TruSeq RNA prep kit. Libraries were then sequenced using an Illumina HiSeq2000 at the Columbia University Genome Center. Multiplex samples with unique barcodes were mixed in each lane, which yielded a targeted number of single-end 100 bp reads for each sample, as a fraction of 180 million reads for the whole lane. RTA software (Illumina, San Diego, CA, USA) was used for base calling, and bcl2fastq (version 1.8.4) was used for converting BCL to fastq format, coupled with adaptor trimming. The reads were mapped to a reference human genome (hg38) using STAR software [38]. RNA-seq data of one patient sample were dropped by the quality control procedure.

### 2.8. Differential Expression Analysis

Differential expression analysis was conducted using DESeq2 [39], an R package based on a negative binomial distribution that models the number of reads from RNA-seq experiments and tests for differential expression. Differentially expressed genes (DEGs) between mutant (DEL) and control (WT) samples were determined statistically with false discovery rate (FDR) correction. A list of significant DEGs was defined by FDR *p*-value < 0.01 with a base mean expression level > 100. The expression changes of the genes within the deletion regions were used as positive controls for experimental validation.

### 2.9. Gene Ontology and Protein–Protein Interaction Network Analysis

To determine if some of the top DEGs share common biological processes, molecular function, or cellular components, the enrichment of Gene Ontology terms was tested using the ToppGene database [40] with default settings. Gene lists of up-regulated and down-regulated DEGs were analyzed separately. To determine the DEGs that physiologically interact with each other at the protein level, we imported the DEG list into the STRING database [41] with default settings. The enrichment of the top protein–protein interaction network was obtained according to the gene ontology enrichment analysis in the STRING package.

### 2.10. Topology Analysis of the Node Contribution to the PPI Network

Centrality measures for protein–protein interaction networks were conducted as described previously in [42]. Briefly, for centrality analysis, 27 different centrality measures were initially selected and used. Principal component analysis (PCA), a linear dimensionality reduction algorithm, was used to determine which centrality measures better determine central nodes within a given network. PCA was done on normalized, computed centrality measures. After the most informative centrality measure was determined, the factoextra package was used to attain an optimal number of clusters. To measure the dissimilarity among clusters, Ward’s minimum variance method was used. To compare the clustering results in the aforementioned PPI network, the Jaccard similarity index was used, relying on the similarity metrics of the clustering results within the BiRewire package. Finally, the score based on the most informative centrality measure was calculated for each node in the network and visualized as a bar plot.

### 2.11. Western Blotting

Total proteins were extracted by using a tissue protein extraction buffer (Thermo Fisher Scientific, Waltham, MA, USA, Ref: 78510) that contained proteinase inhibitor (Complete Mini, Roche, Sigma-Aldrich, St. Louis, MO, USA, Ref: 11836170001) according to the manufacturer’s instruction. Briefly, HBMECs were washed twice with DPBS solution and then incubated in the lysis buffer for 10 min on ice and vortexed vigorously. Samples were spun down at 10,000 g and the supernatant was stored at −80 °C for downstream analysis. Extracted protein was separated on a NuPAGE 4–12% gradient gel (Invitrogen, Thermo Fisher Scientific, Waltham, MA, USA, Ref: NP0322). Primary antibodies (Phospho-CRKL (Tyr207) (E9A1U) Cell Signaling #34940, 1:800; CRKL (D4G7G) Cell Signaling #38710, 1:800) were bound by HRP-conjugated secondary antibodies and visualized using the Azure c600 imaging system. Images were quantified using Image Studio Lite Software (Licor Inc., Lincoln, NE, USA, Ver 5.2), and actin protein (b-Actin AbCam, Waltham, MA, USA # ab227387, 1:6000) was used as an internal input control to normalize all measurements.

### 2.12. Statistical Analysis

For solute permeability, TEER, and ESG (HS intensity) assays, data were presented as mean ± standard error (SE), unless indicated otherwise. The Wilcoxon matched pairs signed-rank test was used for comparisons between control and 22q11.2DS-SCZ data. Kurtosis analysis was used to compare the intensity distribution profiles of junction proteins for the control and 22q11.2DS-SCZ cases. *n* ≥ 6 samples for each case for the solute permeability and TEER experiments, and *n* = 3 samples for each case for junction proteins and ESG quantification. The samples were from at least 3 independent differentiations. For RT-qPCR and Western blot, data were presented as mean ± standard deviation (SD). The Student’s t-test was used for comparison between 22q11.12DS-SCZ and control data. A *p*-value of 0.05 was set as the threshold for statistical significance.

## 3. Results

### 3.1. Blood–Brain Barrier Function is Impaired in the 22q11.2DS-SCZ iBBB

Differentiation of iPSC lines into HBMECs was conducted using established protocols as described previously. We determined the expression pattern of stem cell markers (*OCT4, NANOG*), mesoderm markers (*TBXT*), primary steak markers (*PAX2*), endothelial progenitor markers (*KDR*), endothelial markers (*CD31* and *CDH5*), tight junction markers (*CLDN5* and *OLCN*), and efflux transporters (*ABCB1* and *BCRP*) along with the iPSC differentiation procedure, and confirmed that stage-specific markers are expressed as previously described (Figure A2) [28]. The HBMECs were seeded on the Transwell inserts and cultured until they formed iBBBs. To quantitatively investigate the impaired function of the iBBB formed from the HBMECs derived from the iPSCs of the patients with 22q11.2DS-SCZ, we measured its TEER and permeability (P) to solutes of various sizes. TEER reflects the resistance of the iBBB to ions. We chose sodium fluorescein (NaFl, MW = 376) and Dex-70k as representative small and large solutes because the resistance of different components of the BBB varies distinctively to solutes of these sizes. Figure 1A shows that the TEER of 22q11.2DS-SCZ iBBB (DEL) is 42.6 ± 5.9 Ωcm^2^, about 62% that of the control iBBB (WT), which has a value of 68.3 ± 13.5 Ωcm^2^. Figure 1B,C demonstrates that the permeability to NaFl and Dex-70k is 5.4 ± 1.7 × 10^−6^ cm/s and 0.34 ± 0.11 × 10^−6^ cm/s, respectively, for the 22q11.2DS-SCZ iBBB, which are 1.3-fold and 1.4-fold that of the corresponding control iBBBs. Analysis of the data from individual samples indicated that the alterations were consistent across the three patient–control pairs. The significant decrease (*p* < 0.001) in the TEER and increase (*p* < 0.05) in the permeability confirm the impaired barrier function of the BBB in the patients with 22q11.2DS-SCZ.

### 3.2. Tight Junction Proteins and the Endothelial Surface Glycocalyx (ESG) are Disrupted in 22q11.2DS-SCZ iBBB

The TEER and permeability are determined by the structural components of the BBB. For the iBBB formed by the monolayer of HBMECs, the major structural components are junction proteins (e.g., ZO-1, occludin, claudin-5, VE-cadherin) and the ESG. TEER and permeability to small solutes are mainly determined by the junction proteins while permeability to large solutes is determined mainly by the ESG [25,26,27]. Figure 2 shows the confocal images of ZO-1 and occludin in the 22q11.2DS-SCZ iBBB and the control. Figure A4 in the Appendix B shows those of claudin-5 and VE-cadherin. The right panel in Figure 2A–C shows the comparison of the intensity profiles of ZO-1 and occludin labeling along a ~3 μm line perpendicular to the EC junctions (white lines in the confocal images). The peak intensity of ZO-1 or occludin labeling from the control was used for the normalization. *n* = 720–1080 profiles for 180–270 junctions were determined for each case. The normalized junction protein intensity distribution profiles for the patient iBBB are consistently lower than those for the control iBBB in three paired experiments, indicating deficits in the junction proteins in the BBB of 22q11.2DS-SCZ patients.

Figure 3A–C shows the confocal images of the heparan sulfate (HS) of the ESG at the iBBB of the patients and controls. Comparison of the HS intensity in Figure 3D shows that the ESG in the patient iBBB is only 42% ± 8.3% (range 35–63%) of that of the control. The ESG is also compromised in the BBB of 22q11.2DS-SCZ patients.

### 3.3. Alterations in Focal Adhesion and Vascular Endothelial Growth Factor (VEGF)-Associated Junction Signaling Pathways in the 22q11.2DS-SCZ iBBB

To further determine the molecular mechanisms that contribute to these structural, functional, or cellular deficits in the 22q11.2DS-SCZ iBBB, we conducted RNA sequencing of HBMECs in the iBBB model (Figure 4A). There are 972 differentially expressed genes (DEGs) identified (FDR *p* < 0.01, base mean > 100). We confirmed that the genes within the 22q11.2DS region (i.e., *COMT*, *CRKL*, *HIRA*, *RANBP1*, etc.) were significantly altered and all of them were down-regulated in the HBMECs derived from the patients with 22q11.2DS-SCZ. These down-regulated genes were utilized as positive controls, which helped to provide evidence that the RNA sequencing experiment was indeed successful (Figure 4A and Appendix A). In addition, principal component analysis (PCA) indicated that the first principal component can separate the transcription profiles of the patients from the controls (PC1: 48% variance) (Figure 4B). There are 457 additional DEGs significantly up-regulated (FDR *p* < 0.01, fold change > 2) and 247 down-regulated DEGs (FDR *p* < 0.01, fold change < 0.5) in the patients with 22q11.2DS-SCZ (Figure 4A). Gene Ontology (GO) analysis indicated that the top enriched GO terms for the 457 up-regulated DEGs include “anchoring junction” (GO:0070161, FDR B&H *p* = 2.3 × 10^−36^), “adherens junction” (GO:0005912, FDR B&H *p* = 2.3 × 10^−36^), and “focal adhesion” (GO:0005925, FDR B&H *p* = 8.4 × 10^−35^) (Figure 4C and Appendix A). The top enriched GO terms for the 247 down-regulated DEGs are “cell-cell adhesion via plasma-membrane adhesion molecules” (GO:0098742, FDR B&H *p* = 1.7 × 10^−11^) and “cell junction” (GO:0030054, FDR B&H *p* = 9.2 × 10^−7^) (Figure 4D and Appendix A). These results indicate that the regulation of cell adhesion and junction properties of HBMECs is altered in patients with 22q11.2DS-SCZ. This is consistent with our finding that the barrier function and junction proteins are altered in the iBBB derived from the patients with 22q11.2DS-SCZ.

To further determine which genes within the 22q11.2DS region are involved in the altered permeability of BBB and the related key physiological signaling pathways enriched for other DEGs, we first determined a protein–protein interaction (PPI) network formed by 76 DEGs outside the deletion regions that are enriched for the GO terms of “anchoring junction” and “adherens junction” using STRING (https://string-db.org/, accessed on 18 August 2021). We then analyzed which DEGs within the deletion region interact with the genes within this PPI network (Figure 5A). After that, we quantified the contribution of each DEG within the deletion region to the PPI network by calculating their centrality indices, which are designed to produce a ranking that accurately identifies the most influential nodes in a network. We analyzed 23 centrality index parameters and selected the most informative centrality measures for our PPI network by PCA analysis (Figure A3). We used “diffusion degree score”, the most informative centrality parameter to determine the relative contribution of each node in the network. The topology analysis of the PPI network indicated that CRKL, which is within the 22q11.2 deletion region and encodes an adaptor protein, connects with multiple nodes in the protein–protein interaction network and had a much stronger contribution to the network compared to the other genes within the deletion regions (Figure 5B), suggesting that it may play a critical role in cell adhesion and cell junction functions. We further annotated the final PPI network with the expression fold change (DEL/WT) using RNA-seq data for each node (gene) (Figure 5A). The log2 expression fold change of each gene is shown by a red color, indicating up-regulation, and a green color, indicating down-regulation, in 22q11.2DS-SCZ. Of note, *claudin-5* [12] and *TBX1* [11,43] are two other genes within the 22q11.2 microdeletion region that have been previously shown to play a role in vascular/BBB development. However, in our model, the RNA expression levels of both of these genes are very low, suggesting that these two genes from within the deleted region have a minor contribution to the permeability changes observed at the early HBMEC developmental stage (Appendix A). In contrast, the CRKL gene is highly expressed in HBMECs. We confirmed that both the CRKL protein and its active form phosphor-CRKL (Tyr207) are decreased by 50% in the HBMECs derived from patients with 22q11.2DS-SCZ compared to the controls (Figure 6A). The Abl kinase signaling is important in regulating VEGF-mediated endothelial barrier function, and CRKL is the key downstream effector of Abl [43]. Moreover, CRKL is an adaptor protein that controls a set of small GTPases, such as RAC1 and RAP1, which are critical in regulating cell adhesion and junction stability [43,44]. VEGF is known to be a dosage-sensitive gene required for normal development, and VEGFA164 isoform deficiency results in birth defects in mice reminiscent of those found in patients with 22q11.2DS [45]. Therefore, it is likely that decreased *CRKL* expression is associated with or contributes to the increased BBB permeability in patients with 22q11.2DS-SCZ.

We further confirmed the alterations of several other key nodes in the PPI network using RT-qPCR. Besides a decrease in the CRKL RNA, using RT-qPCR, we confirmed that *FLT1/VEGFR1* expression is significantly up-regulated (*p* = 0.038, *n* = 6) and *KDR/VEGFR2* is significantly down-regulated in 22q11.2DS-SCZ (*p* = 0.03, *n* = 6) (Figure 6B). These results suggest an alteration in the VEGF signaling pathway in the pathophysiology of 22q11.2-associated BBB permeability deficits. It has been shown that FLT1/VEGFR1 physically interacts with CRKL [46]. In addition, the up-regulation of *EPAS1/HIF-2a*, a basic helix–loop–helix/PAS domain transcription factor, is also confirmed. *EPAS1/HIF-2a* is expressed preferentially in vascular ECs and involved in the physiological response of ECs to hypoxia [47], and the VEGF receptors are positively regulated by EPAS1/HIF-2a. Interestingly, both *EPAS1/HIF-2a* and its regulatory targets, *FLT1*, *PHLDA1*, and *DUSP5*, are also significantly up-regulated in the HBMEC transcriptome derived from patients with 22q11.2DS-SCZ (Appendix A). We further confirmed these changes by RT-qPCR (Figure 6B). In fact, both the PHLDA1 and DUSP proteins have been shown to be negative regulators of the VEGFR2 signaling pathway (Figure 6C) [48,49]. Together, these results suggest that haploinsufficiency of *CRKL* might be associated with alterations in the VEGF receptors and their downstream signaling in 22q11.2DS-SCZ HBMECs (Figure 6C). The disruption in the CRKL/FLT1 interaction could be one of the key mechanisms underlying the deficits observed in the 22q11.2DS iBBB junction barriers.

## 4. Discussion

Our results for the direct comparison in the structure and function of the 22q11.2DS-SCZ and control iBBBs clearly show that the barrier integrity of the BBB is compromised in patients with 22q11.2DS-SCZ. An investigation of junction proteins and ESG expression levels by immunostaining indicated alterations in the tight junction protein deposit and ESG expression. Further RNA sequencing revealed the transcriptomic changes and identified *CRKL* as an important candidate gene responsible for the compromised BBB in patients with 22q11.2DS-SCZ. Our results suggest that CRKL could be a key signaling network downstream of VEGF that is altered in HBMECs derived from patients with 22q11.2DS-SCZ. We further confirmed the changes in the key components of this signaling network using RT-qPCR and Western blotting. In a future study, we can also utilize a Western blotting method to further quantify the junction proteins and a liquid chromatography (LC)–mass spectrometry (MS) method to quantify the ESG, as described in [50].

One concern of this study is that the iBBB formed on the Transwell filter from the HBMECs derived from iPSCs using the current method [28,29,30] may not be the most biomimetic, since its permeability is not as low as measured in the in vivo experiment. It is worthy to note that various parameters, including cell culture conditions, measurement techniques, and analysis methodology can produce a sizeable difference in TEER values. Lippmann et al. reported that adding 10 μM RA (retinoic acid) greatly enhanced TEER from 73.2 to 1920 Ω cm^2^ for the iBBB generated from their iPSC-derived HBMECs [51]. We did not have 10 μM RA in our cell culture medium as we used a no-RA differentiation scheme described in [29,30]. In their protocol, they mentioned that if an experiment may be negatively impacted by RA addition, HBMECs can also be successfully differentiated without RA supplementation on day 6 and day 8 during subculture. They also indicated that if no RA is added on day 6 and day 8 of differentiation (no-RA differentiation scheme), no RA should be added to the medium for subculturing the HBMECs on the Transwell filter. For this reason, we did not add 10 μM RA when generating iBBBs from our iPSC-derived HBMECs. To judge if an in vitro model is good enough, we should compare it with the in vivo data. Compared to our in vivo permeability measurement for rat cerebral microvessels [27,52], the permeability of the control iBBB to sodium fluorescein, 3.8 ± 1.0 (SD) × 10^−6^ cm/s, and to FITC-Dex-70k, 0.24 ± 0.06 (SD) × 10^−6^ cm/s, are twice and 1.6-fold that of the in vivo data, respectively. Our current iBBB model has indeed higher but comparable permeability to that of rat cerebral microvessels. One reason is that we did not have other cell types such as astrocytes and pericytes in generating the iBBB. These components should reduce the permeability of the current simplified iBBB with only HBMECs, and they also contribute to the altered BBB in 22q11.2DS-SCZ patients. Another reason is that we may need to optimize the differentiation and subculturing protocols for our iPSCs to generate iBBBs with improved junction barrier functions. Growing iBBBs under flow conditions in microfluidic chambers or channels can improve the iBBB barrier function with lower permeability and higher TEER [53,54]. In the future, we will construct a more biomimetic iBBB in 3D with a circular cross-section, and co-culture HBMECs with astrocytes, pericytes, and neurons in a microfluidic system to better investigate the BBB in 22q11.2DS-SCZ patients.

Another concern is that we did not observe high expression of tight junction protein, claudin-5, and adherens junction protein, VE-cadherin in the iBBB from either 22q11.2DS-SCZ patients or controls according to the RNA-seq and immunostaining results. Figure A4 demonstrates that although claudin-5 and VE-cadherin have a significant expression on the iBBB, they are not localized at the junctions as ZO-1 and occludin (Figure 2). There are several explanations for such discrepancies. First, claudin-5 expression is sensitive to RA treatment. Our time course RT-qPCR showed that claudin-5 was expressed in all the stages of HBMEC development but dramatically decreased at the terminal HBMEC stage (Figure A2). This observation is consistent with what had been described in the original paper [51]. Second, claudin-5 is one of the densely expressed tight junction proteins in the BBB [55,56,57] and the claudin-5 gene is within the 22q11.2 microdeletion region [58,59]. Crockett et al. [60] assessed claudin-5 expression by Western blotting in their iBBB and found no change in the claudin-5 protein level in the iBBB between 22q11.2DS patients and controls. All these results suggest that claudin-5 gene expression is not a dominant contributing factor in the compromised barrier integrity of 22q11.2DS at these early neurodevelopment stages. Therefore, there must be other factors (i.e., CRKL as suggested by our results) that contribute to the observed permeability changes. Nevertheless, we did see significantly lower expression levels of other tight junction proteins, ZO-1 and occludin at cell–cell junctions that were consistently associated with the increased permeability in the iBBB of 22q11.12DS-SCZ. Third, Lu et al. showed that the HBMECs obtained with the approach used by Lippmann et al. are more likely to have neuro-ectodermal epithelial cell features instead of endothelial cell features [51,61]. Indeed, low *CLDN5* and *CD31* expressions were observed in our terminal HBMEC transcriptome, which are consistent with Lu et al. [61]. On the other hand, our time course RT-qPCR of key mesoderm and endothelial cell markers suggested that *CD31* and claudin-5 mRNA were expressed at all differentiation stages, but claudin-5 was reduced in the terminal HBMEC stage. We speculated that RA treatment, while enhancing barrier function, might have a negative impact on the endothelial cell features of HBMECs. This speculation needs further investigation by using a stage-specific single-cell RNA-seq approach. Regardless of the cell type features, the compromised barrier function is obvious in 22q11.2DS-SCZ.

To investigate the molecular mechanism associated with the compromised barrier integrity of the BBB in 22q11.2DS-SCZ, we conducted RNA sequencing of HBMECs from the iBBB model. We found that genes governing the biological processes related to blood vessel morphogenesis and cell–cell junctions are significantly altered in HBMECs derived from patients with 22q11.2DS-SCZ. To further determine the key physiological signaling pathways among these altered genes, we analyzed the protein–protein interaction network and identified *CRKL*, an altered gene within the 3M 22q11.2 microdeletion region, which is highly expressed in the control HBMECs and significantly decreased in HBMECs derived from patients with 22q11.2DS-SCZ. CRKL-associated tyrosine kinase signaling has been shown to be critical for the VEGF-mediated permeability changes in endothelial cells [43]. Previous studies (including ours) have shown that VEGFR2/KDR signaling is directly involved in the regulation of junction proteins such as ZO-1, occludin, and VE-cadherin, which modulate the BBB permeability [62,63]. In addition, VEGFR2 signaling contributes to glycosaminoglycan regulation by VEGFA and VEGFC in the endothelial glycocalyx [64]. The endothelial glycocalyx is a barrier between the circulating blood cells and the ECs forming the vessel wall, which partially restricts the passage of large molecules, e.g., plasma proteins and cytokines, across the BBB [14,19]. Our finding that *CRKL* is expressed in HBMECs and compromised in the 22q11.2DS HBMECs implies that CRKL-associated signaling pathways play an important role in permeability changes in the iBBB derived from patients with 22q11.2DS-SCZ. The association of the *CRKL* gene with the altered junction and adhesion network might provide a novel insight on understanding BBB deficits in the patients with 22q11.2DS-SCZ. Although from the network and expression pattern analysis we favor the contribution from *CRKL*, there are over 70 genes compromised in the 3M deletion regions in 22q11DS; thus, we could not rule out the impact from other genes. In fact, it is likely that multiple genes could change the status of cells. For example, there are at least six genes within the deletion regions (*PRODH*, *MRPL40*, *TANGO2*, *ZDHHC8*, *SLC25A1*, and *TXNRD2*) that are implicated in mitochondrial functions, which might lead to hypoxia status and alterations in *EPAS1/HIF-2a* (see [65]). A further investigation is needed to test these possibilities.

Last but not the least, our results suggested some intriguing connections between HBMEC signaling and neurodevelopment in terms of “neurovascular coupling” [66]. Our GO analysis suggested down-regulation of genes in synaptic formation in addition to alterations in cell–cell junction and cell adhesion structures. This result seems to support our hypothesis that HBMEC/BBB-related deficits in 22q11.2DS contribute to its much higher risk for developing neurodevelopmental or neuropsychiatric disorders, such as schizophrenia and autism [5,67]. In this regard, our iPSC-derived BBB model provides an initial clue on the potential connection between BBB and neuronal phenotypes, which is hard to be investigated in the developing human brain. A further multiple-stage single-cell RNA-seq analysis and morphological characterization on this platform would provide more information on dissecting vascular contributing factors of neuropsychiatric disorders in 22q11.2DS.

## 5. Conclusions

Our study on the function and structure of an in vitro BBB model indicates that the integrity of the BBB is compromised in patients with 22q11.2DS-SCZ. A transcriptome study further suggests that the transcriptional network related to cell–cell junctions and cell adhesion in the compromised BBB is substantially altered. Among them, *CRKL*, a gene within the 22q11.2 microdeletion region, is significantly decreased and strongly associated with the changes of this transcriptional network.

## Figures and Tables

**Figure 1 cells-10-02576-f001:**
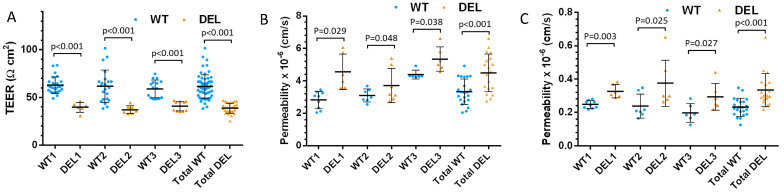
Comparison of barrier function of the iBBB between control (WT) and 22q11.2DS-SCZ (DEL). TEER (**A**), Permeability to a small solute, NaFl (**B**), and to a large solute, Dex-70K (**C**), for 3 paired control and 22q11.2DS-SCZ samples. Values are mean ± SD. *n* ≥ 6 for each case were from at least 3 independent experiments (differentiations).

**Figure 2 cells-10-02576-f002:**

Comparison of junction proteins in the iBBB between control (WT) and 22q11.2DS-SCZ (DEL). Confocal images showing tight junction proteins (ZO-1 and occludin) at the iBBB for the control (left panel) and 22q11.2DS-SCZ (middle panel) from 3 paired iPSCs (**A**–**C**). The right panel shows the comparison of the normalized intensity profiles of ZO-1 and occludin labeling along a ~3 μm line perpendicular to the EC junctions (white lines in the confocal images). *n* = 3 samples with 180–270 junctions (720–1080 perpendicular lines) analyzed for each case. * *p* < 0.05.

**Figure 3 cells-10-02576-f003:**
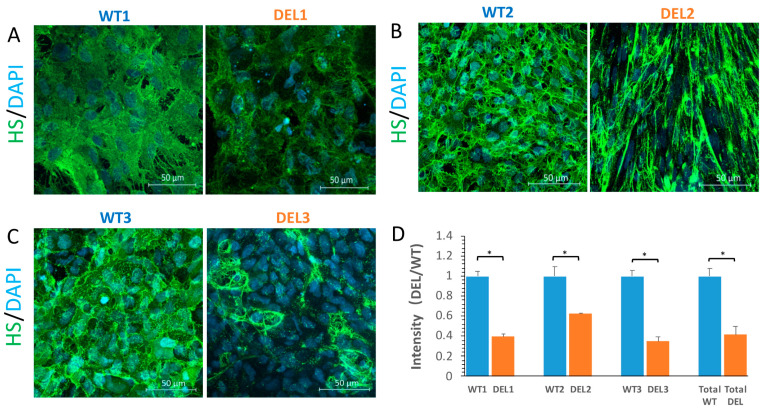
Comparison of the ESG at the iBBB between control (WT) and 22q11.2DS-SCZ (DEL). Confocal images showing heparan sulfate (HS) of the ESG at the iBBB for the control (left panel) and 22q11.2DS-SCZ (right panel) from 3 paired iPSCs (**A**–**C**). (**D**) Normalized HS intensity at the iBBB for the control and 22q11.2DS-SCZ. *n* = 3 samples with 9 fields (each field 320 μm × 320 μm) analyzed for each case. * *p* < 0.05.

**Figure 4 cells-10-02576-f004:**
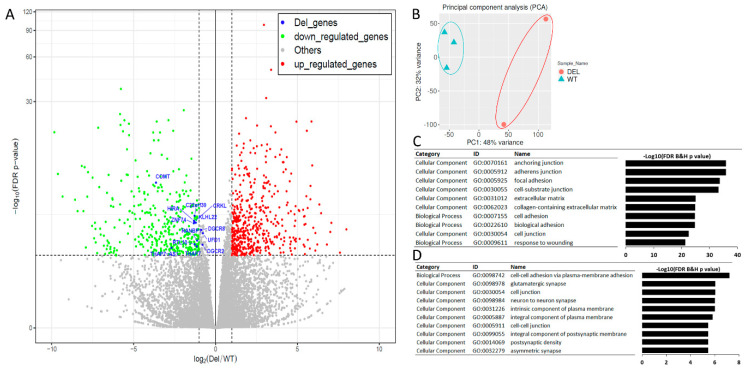
RNA-seq analysis of the iBBB model of 22q11.2DS-SCZ. (**A**) Volcano plot of DEGs. The blue spots indicate the genes within the 22q11.2DS deletion region. The red spots indicate the 457 up-regulated genes. The green spots indicate the 247 down-regulated genes. (**B**) PCA analysis indicates that the transcriptome profile can separate patients from controls with the first principal component (PC1). DEL, patient with 22q11.2DS-SCZ; WT, normal control. (**C**) GO analysis of up-regulated DEGs indicates that GO terms related to junction and cell adhesions are significantly altered in 22q11.2DS-SCZ HBMECs. (**D**) GO analysis of down-regulated DEGs indicates that GO terms related to junction and cell adhesions are significantly altered in 22q11.2DS-SCZ HBMECs (Fisher’s exact test).

**Figure 5 cells-10-02576-f005:**
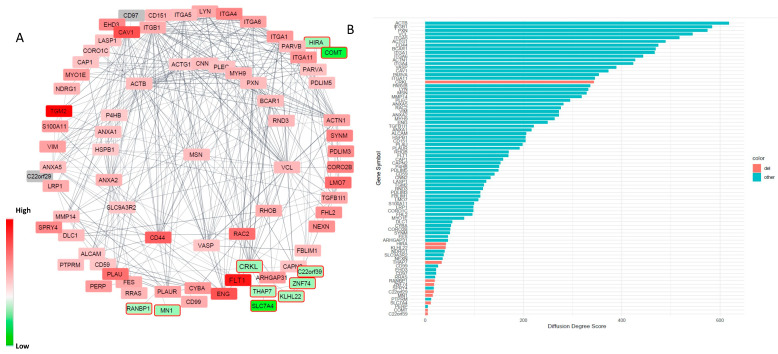
Alterations in the cell junction and adhesion network. (**A**) An altered PPI network involved in cell junction and adhesion functions. Red-dotted rectangle indicates the interaction between CRKL and VEGR1/FLT1. (**B**) The centrality measure analysis of the contribution of each node to the PPI network. The x-axis is the “diffusion degree score” (see the method) and the y-axis is the nodes (gene symbols) ranked according to their diffusion degree score.

**Figure 6 cells-10-02576-f006:**
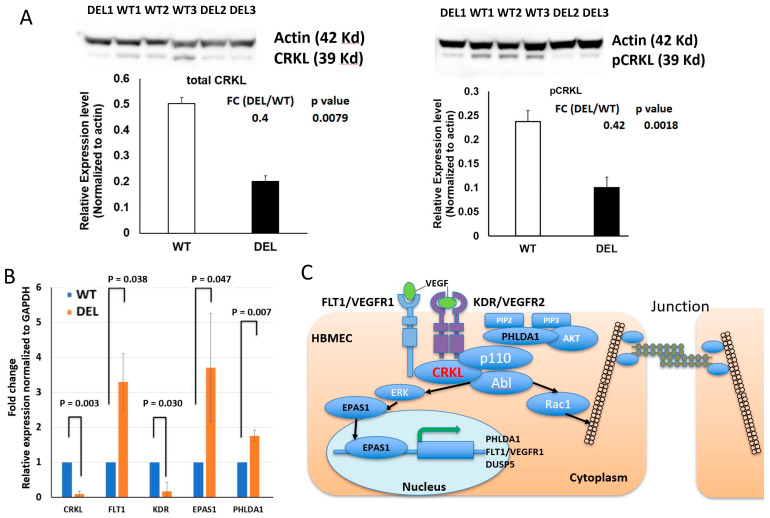
Involvement of CRKL and the VEGF signaling pathway. (**A**) Western blot confirmed decreased expressions of CRKL and pCRKL in 22q11.2DS-SCZ (Student’s t-test *p* = 0.0079 for total CRKL and *p* = 0.0018 for pCRKL). (**B**) RT-qPCR confirmed alterations in CRKL (Student’s t-test *p* = 0.003), VEGFR1/FLT1 (*p* = 0.038), VEGFR2/KDR (*p* = 0.03), EPAS1/HIF-2a (*p* = 0.047), and PHLDA1 (*p* = 0.007) in 22q11.2DS-SCZ (sample size = 6, technical repeats = 5). (**C**). A working model of how alterations in CRKL-VEGF receptor interactions lead to deficits in the junction barriers. Ncase = 3, Ncontrol = 3, and error bar = standard deviation for all analyses.

## Data Availability

Data is contained within the article or Appendix A.

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
