# Peer review of "Investigation of Neurodevelopmental Deficits of 22 q11.2 Deletion Syndrome with a Patient-iPSC-Derived Blood–Brain Barrier Model"

_cells, 2021, doi:10.3390/cells10102576_

Round 1

Reviewer 1 Report

In this work Li Y and colleagues investigated the effect of 22q11.2 deletion (ethiology for DiGeorge syndrome) on iPSC-derived endothelial cells phenotype.  The authors 1) found changes on the permeability to small and large solutes of a iBBB composed of endothelial cells containing the abovementioned deletion, 2) observed changes in the amount of ZO-1 and Occludin present in confluent healthy and diseased cells, 3) conducted RNAseq experiments to determine the genes with altered expression, and 4) proposed a working model to explain the relation between gene expression and the outcomes observed at the beginning of the work.

Overall the manuscript is well written and organized, and most of the images used are clear and legible.  The science and experimental strategy used to support and test the hypothesis are solid and support most of the authors conclusions.  Nevertheless, I believe some changes to the manuscript and complementary experiments are needed to confirm the obtained results.  As such, I propose major revisions to this manuscript.  Please consider the following:

Abstract

  1. No direct reference to the name “DiGeorge Syndrome” is done in the abstract. Please consider adding this.
  2. Consider highlighting better the novelty and impact of the results in the abstract.

Introduction

  1. Consider moving the phrase starting with “We found…” in lines 81 and 82 to the results and discussion section.

Materials and Methods

  1. In section 2.1 please evidence which of 3 deletions studied (del1, del2, del3) correspond to the hiPSCs cell lines with the 22q11.2DS deletion and the modified hiPSC.
  2. Please do a re-check on the abbreviations used (example: RA in line 125).
  3. In section 2.3 it is mentioned that cells took 6-7 days to reach confluency to form the iBBB. How was this monitored? Microscope? Impedance? Indirect metabolic assay (e.g. Alamar Blue® or Presto Blue®)? Please add this information here, and if possible corresponding supporting data.
  4. In section 2.5, please provide more information about the antibodies used (host, clonality and reference).
  5. I could not find Table S1 with the information of the antibodies used for Western-Blot (WB) in the documents provided. Please address this.
  6. In section 2.6, what was the specific protocol used for fluorescence quantification? Were the images processed equally?
  7. Please do a re-check for minor typing errors. For example, in line 232 “protein inhibitor” should read “proteinase inhibitor”.

Results

  1. Regarding Figure 1, the n=6 used for each WT/Del sample is composed of independent experiments or technical replicates?
  2. Figure 2 contains the first set of fluorescence quantification data. While this method is good for assessing overall presence of the proteins in the cells, it should be complemented with a more precise quantification method. Please consider doing WB or another appropriate technique.
  3. Is there a reference for the claim done in lines 300 and 301?
  4. Figure 3 contains the second set of fluorescence quantification data. While this method is good for assessing overall presence of HS in the cells, it should be complemented with a more precise quantification method. If possible, consider quantifying HS with another method., such as the LC-MS/MS method described by Li G and colleague, 2015 (10.1021/acschembio.5b00011).
  5. In lines 311-315 the authors mentioned the down- and up-regulated genes found in the RNAseq experiment. Can you provide some examples in the text?
  6. Please comment on the results of Figure 4D about cellular components found in neurons that are expressed by the HBMECs. Consider adding this information to the discussion section.
  7. Please consider adding HIF-2a as an alternative name for EPAS1.
  8. Please consider reforming the strong claim presented in lines 399 to 401. In this work the authors do not selectively inhibit or rescue the pathway/CRKL to fully confirm the relationship proposed.
  9. In figure 6, please consider adding WB results for EPAS1, PHLDA1, FLT1 and KDR.

Discussion

  1. The qPCR data presented in figure 6 reveals an intriguing relationship between CRKL and EPAS1. Please consider discussing the role of other molecular pathways (e.g. FGF2 Tyrosine-Kinase Receptor, hypoxia, glucose, Akt and NF-kB) in leading to an increase in EPAS1 despite a reduction in CRKL levels reported.
  2. The authors report that the iBB model used has different permeability values compared to the previous in vivo results found. Please comment on the simplicity of the model used (absence of pericytes and glia cells) and possible limitations to understanding all the changes occurring in the BBB of DiGeorge Syndrome patients.
  3. Please comment on the usefulness of more complex models and/or tissue engineering constructs to better study the BBB function.
  4. Please pay attention to the phrase “Our finding that CRKL is highly expressed in HBMECs…” (line 441). CRKL is not highly expressed in normal HBMECs. Consider replacing “highly” by “physiologically”.

Appendix

  1. Figures A1D, A2 and A3 have low resolution. Please consider replacing by high-resolution versions.

Reviewer 2 Report

The authors examine the BBB properties of brain endothelial cells (BEC) derived from iPSCs from healthy and 22q11.2 Deletion Syndrome. They describe a decrease in BBB permeability and integrity in patient BECs and transcriptomic analysis reflects this phenotype in altered cell-to-cell junction expression profiles. Although this paper aims to assess the role of disrupted BBB as a contributor to the neurodevelopmental phenotypes of the afflicted patients, the data should be strengthened as discussed below:

 The authors presented the TEER data as a ratio of Del/Wt - absolute TEER values would be more informative especially if the purpose it to highlight changes in barrier integrity. The protocol used herein (Qian et al), described TEER values for control iPSC lines around 3000 ohms.cm2. The control TEER values in this manuscript are 68 ohms.cm2 - this is very low for this well described and reproducible protocol which suggests that the differentiation may need to be optimized for the iPSC lines - particularly for the control to be able to confidently discriminate barrier integrity vs suboptimal differentiation in patient cells. Further to this point – the wildtype and del iPSC lines proliferate the same? The seeding density of iPSCs is critical to BBB formation. If the growth rate is different the differentiation would be skewed – it would be important to try different seeding densities for the wildtype to improve over TEER values.

Figure 2 – Claudin 5 density expression levels and cellular distribution (membrane vs cytoplasmic) should be added to the panel as it has a critical role in BBB-related barrier formation. The fact that the authors did not see Claudin 5, or VE-cadherin or CD31 (as it is lacking) is concerning to the endothelial phenotype of these cells. Recent evidence suggests that the BEC derived using the established differentiation protocols are not endothelial but rather neuroectodermal epithelial cells. These cells have been shown to lack Claudin 5, VE-Cadherin and CD31 in which case the BBB link needs to be reconsidered and re-evaluated in this paper. If these cells are indeed neuroetodermal epithelial cells, which lack functional attributes of endothelial cells, what is the significance of these findings? This needs to be discussed.

More over, are the BECs in this study responsive to VEGF and able to undergo angiogenesis? Does adding VEGF compromise TEER in these cells? VEGF signalling is among the pathways affected in the iPSC-derived BBB differentiation protocol. In this scenario – what would the significance of the involvement of CRKL in VEGF signaling pathway be? More specifically – would this be more neurodevelopmental or neuropsychiatric related rather than BBB related?

Given this data – the authors need to mine the RNASeq data, in reference to the Lis lab, to more accurately identify the cell type they have derived in this differentiation. TEER is low and key BBB and endothelial requisite markers missing. It would be advisable to re-examine the data in context of the neuroectodermal epithelial cells and how this may contribute to neurodevelopmental disorders of 22q11.2 Deletion Syndrome.

To the last point, the transcriptomics data is a valuable dataset – more can be dissected and distilled out of this data set with a more focused look at the neuroectodermal epithlium with the neurodevelopmental phenotype would significantly strengthen/refocus this paper.

On line 121, they said “filtered cell through 3uM membrane”,--3um is really small for cells, typical iBEC is 10-12 uM, the authors should use 40uM filter. Is this a typo?

The discussion was very brief – highlighting key deficits identified in the BBB phenotype but no real consequence discussion on how that may affect the BBB-centric data/interpretation. Significant revisions, with a refocus on the RNASeq data, can potentially bring meaning to this manuscript.

Reviewer 3 Report

Li et al developed a personalized blood-brain barrier model to investigate the neurodevelopmental deficits of 22q11.2 deletion syndrome. In particular, the authors investigated if 22q11.2 deletion syndrome compromise the BBB integrity using endothelial cells derived from patients specific iPSCs lines. Authors found the compromise in BBB integrity and TEER developed using 22q11.2 deletion syndrome-associated Schizophrenia patients. This change was correlated with the expression of junction proteins and glycocalyx. Authors also found a particular gene CRKL, within 22q11.2 microdeletion region which dictates the transcriptional changes observed.

This article interests broad audience in the field neurodevelopmental disorders and open doors to explore personalized medicine.

Minor Issues:

  1. Section 2.7: How were the cells recovered from the transwells for RNA sequencing.
  2. Figure 2B: Del2, Occludin/DAPI image shows a pattern of cell alignment. Was this observed in all the images? If not, please replace the image.
  3. Figure 3D: Please improve image resolution.
  4. Supplementary Figure A1D and A3 needs better resolution images
  5. Authors should improve the discussion of the results especially the RNA sequencing results.

Reviewer 4 Report

Li, Xia, Zhu et al. provide a very nice presentation of an iPSC-derived BBB model to compare differences between diseased and healthy patient population. Authors show, that HBMEC junctional expression intensity decreases in 22q11.2 Deletion Syndrome patients leading to lower TEER and higher permeability in culture. One of the main glycocalyx components are also involved in the barrier integrity impairment. Authors describe the decreasing expression of the protein CRKL, which could be a main component behind the junctional impairment through the VEGF pathway. 

Technical questions: 
- Why HBMECs were filtered through a 3um membrane after Accutase detachment and before seeding to Transwell inserts? This seems to be rather a small pore size for a HBMEC to be filtered through. (Usually 20-40um mesh is used for such purpose.)
- Were cells cultured on a growth factor reduced Matrigel? Please indicate. 

Methods/Results: image analysis
In my opinion for the image analysis of junctional proteins 3 random fields/sample is not enough to reach correct statistical power (Figure 2). I recommend using at least 7-8 random spots to perform the image intensity analysis. Please provide more parallels in this analysis, and explain in the Methods section how exactly the ImageJ analysis was performed. 
During analysis of fluorescent intensity of junctional protein expression total intensity measurement of the staining is not very acceptable due to the obvious cytoplasmatic staining in the figures presented. 
The same is not valid for the HS staining, there total intensity can be used, but n number for the random fields/sample has to be elevated as well. 

Discussion:
One of the things missing from the discussion is what kind of gene expression changes were found for ZO-1 and occuldin and HS during the study? Expression level changes are mentioned, but it is not evident for the reader, whether authors refer to the fluorescent intensity differences found earlier or to gene expression differences. Please clarify. Also discuss a bit glycocalyx component changes. 

Appendix: Figure A2
Why is CLDN5, OCLN, BCRP and ABCB1 at the same expression level from the first day of the iPSC differentiation process to the 12th day? Stem cell markers, such as OCT4 or NANOG go down with time, but these brain endothelial specific markers stay very similar. 

Comment to Figure legends: 
Please provide n numbers and the exact statistical analysis performed at each figure's legend. 

Stilistic and Figure formatting comments:

Figure 1 - graph legend could be reverse, since the order in the graph is WT, Del, not Del, WT. And legend could be moved to the left top corner of the graph to be more visible (like in Figure 6B).

Figure 3D is blurry. 
Figure A1D is not readable at all. 
Figure A3 is poor quality. 
line 21 - junction proteins - junctional proteins
line 24 - junctionS
line 254 - determineD
line 259 - cultureD 
etc.

After providing answers to my questions and more parallels in the image analysis experiments I find this manuscript acceptable to the current journal. 

Round 2

Reviewer 1 Report

The authors have addressed all my concerns and improved the quality of the manuscript, and for that I congratulate them. The paper can be accepted without further improvements.

Final note: Regarding answer 12 (and partly answer 14), I believe the authors presented sufficient evidence to support functional changes in barrier function of their iBBB model (TEER and molecular studies), complementary to the IF intensity analysis performed.  While the method can be considered standard for the iBBB model presented, as evidenced by previous published work, I suggest the authors to consider complementary WB analysis in future works.

Reviewer 2 Report

The reviewer thanks the authors for their responses but still feels that the manuscript still requires revisions.

In response to question 1 - the authors stated that they intentionally did not add RA as part of their differentiation protocol, resulting in low suboptimal barrier forming TEER values. Can the authors elaborate on their rationale for this otherwise routine protocol change. Especially if the purpose of the manuscript is to examine barrier properties, which RA supplementation known to induce bringing the experimental models "closer" to what is observed in vivo? RA-treatment has also been shown to induce more membrane specific CLN5 expression that can be more information in dissecting alterations on the TJ and functional barrier properties. I would urge the authors to include data on RA vs NON-RA treated BEC phenotypes in this manuscript to better understand the effect on barrier properties.
